# The Influence of SARS-CoV-2 Pandemic in the Diagnosis and Treatment of Cervical Dysplasia

**DOI:** 10.3390/medicina57101101

**Published:** 2021-10-14

**Authors:** Anca-Maria Istrate-Ofițeru, Elena-Iuliana-Anamaria Berbecaru, Dan Ruican, Rodica Daniela Nagy, Cătălina Rămescu, Gabriela-Camelia Roșu, Larisa Iovan, Laurențiu Mihai Dîră, George-Lucian Zorilă, Maria-Loredana Țieranu, Dominic-Gabriel Iliescu

**Affiliations:** 1Department of Histology, University of Medicine and Pharmacy Craiova, 200349 Craiova, Romania; ancaofiteru92@yahoo.com (A.-M.I.-O.); nicola_camelia92@yahoo.com (G.-C.R.); iovan.larisa@gmail.com (L.I.); 2Department of Obstetrics and Gynecology, Emergency County Hospital of Craiova, 200642 Craiova, Romania; iuliaberbecaru@gmail.com (E.-I.-A.B.); ruican.dan@hotmail.com (D.R.); rodica.nagy25@gmail.com (R.D.N.); catalina.ramescu@yahoo.com (C.R.); laurentiu.dira@yahoo.com (L.M.D.); zorilalucian@gmail.com (G.-L.Z.); dominic.iliescu@yahoo.com (D.-G.I.)

**Keywords:** SARS-CoV-2, dysplasia, colposcopy, biopsy, morphopathology

## Abstract

*Background and objectives*. The risk of developing invasive cancer increased during the COVID-19 pandemic, especially in Romania, where the incidence of this disease is high due to limited medical education and broad screening. This study’s objective is to analyze the number of patients admitted with different types of cervical dysplasia and the treatment applied for the lesions during the SARS-CoV-2 pandemic compared to the same period for the year before the pandemic. *Materials and methods*: This is a retrospective study that took place in the Obstetrics and Gynecology Clinics I/II (OG I/II) of the Emergency County Hospital of Craiova during the SARS-CoV-2 pandemic (SP) (15.03.2020–14.03.2021) and in the 12 months before (non-pandemic period) (NPP) (15.03.2019–14.03.2020). The study includes 396 patients with pathological PAP smear results. All the patients included in this study were clinically examined and with colposcopy. The patients with Low-Grade Dysplasia were managed in a conservatory manner and reevaluated after six months. The patients with High-Grade Dysplasia were admitted for an excisional biopsy of the lesion. The excised fragments were sent to the Pathological Anatomy Laboratory for a histopathological examination. *Results*: This study reveals a decrease of more than half in the number of patients admitted with cervical intraepithelial neoplasia (CIN) lesions during the pandemic compared to the same period of the year before. The number of biopsies and excisional procedures has been decreasing by more than a factor of three during the pandemic period compared to the year before. *Conclusion*: During the SARS-CoV-2 pandemic, we found that the patients’ admission rate, diagnosis, and treatment was almost four times lower. As hospital restrictions were not dictated for cancer/precancer management during SP, we may assume that the differences were due to the fear of becoming infected with SARS-CoV-2 due to hospitalization. In the context of poor screening performance and high cervical cancer incidence, the influence of the SP may result in a further increase of severe cases related to this condition.

## 1. Introduction

Cervical cancer is the third most common type of gynecological malignancy in developing countries [1,2]. After treating the dysplastic lesions, these women have been reported to have a higher risk of invasive cervical cancer than the general population for several years after treatment [3].

Worldwide, this type continues to be one of the most common cancers among females, being the fourth most common after breast, colorectal, and lung cancer. In 2018, it was estimated that there were approximately 569,847 new cases of cervical cancer [4]. Studies have shown that screening tests for this condition should reduce the incidence of cervical cancer by up to 80% [5].

Cervical cancer is one of the malignant pathologies that can be prevented by screening and early detection. The situation in Romania is quite alarming regarding this pathology. According to the International Agency for Research on Cancer, in 2020, Romania occupied the second position in Europe (EU) after Montenegro [6]. Cancer mortality rates are the second most common cause of death in Romania, with an estimated incidence of 148.84 deaths per 100,000 people. This statistic ranks our country 20th worldwide. The most frequent malignancies that increase mortality in Romania are lung neoplasia, breast, colon, prostate, and uterine cervical cancers [7]. The healthcare system in our country needs multiple improvements, similar to other developing countries in the European Union (EU). There are strong differences between member countries of the EU regarding cervical cancer mortality. These differences are determined by the implementation of the screening [8]. The average mortality rate from cervical cancer in the EU is 3.4/100,000 women; however, in Romania, the rate is higher, above 11/100,000 women [9].

The most used screening method for cervical cancer is cervicovaginal cytology (Pap smear). It is performed with a certain periodicity depending on the age and pathological history of the patients. According to the American College of Obstetricians and Gynecologists (ACOG) and the WHO, a Pap smear should be performed every three to five years after the beginning of sexual activity [10,11].

When a Pap test revealed a high-grade squamous intraepithelial lesion (HSIL), the patients were referred for colposcopy and biopsy.

The American Society of Cervical Cancer and Pathology (ASCCP) recommends treating the HSIL-like intraepithelial cervical lesions 2, 3 (CIN 2, 3). The CIN 2, CIN 3, or CIN 2/3, with a suggestive colposcopy for HSIL, is better to be excised, except for the lesions diagnosed in pregnant or young women [9,12].

In December 2019, the new coronavirus of severe acute respiratory syndrome (SARS-CoV-2) appeared unexpectedly in Wuhan, China. Since then, it has spread globally, [13,14], leading the WHO to announce a global pandemic [15].

To prevent the spread of COVID-19 and optimize the necessary medical resources in emergency areas [16,17,18], the WHO recommended the cancellation of elective surgeries in hospitals [19,20]. Protocols have been adopted to provide safety for patients and doctors, continuing to restrict surgical interventions to emergencies only [21,22,23].

However, the need for care in gynecological disorders continues, as well as the need for adapted, special public health measures to reduce the level of contagion during public healthcare. The reduced addressability of gynecological pathology during the pandemic is known. The extent and potential consequences of preneoplastic lesions of the cervix must be quantified and made known, and this is the main objective of our study.

The aim of our study is to analyze the number of patients admitted with different types of cervical dysplasia and the treatment applied for the lesions during the SARS-CoV-2 pandemic compared to the same period of the year before the pandemic.

## 2. Materials and Methods

This study presents a retrospective analysis of the cervical precancerous lesions managed in the Obstetrics and Gynecology Clinics I/II (OG I/II) of the University Emergency County Hospital of Craiova during the first year of the SARS-CoV-2 pandemic (SP) (15.03.2020–14.03.2021) and in the 12 months before (non-pandemic period) (NPP) (15.03.2019–14.03.2020). The patients included in this study were clinically examined and, due to pathological PAP smear results, they underwent a colposcopic examination. Using this technique, cervical lesions can be highlighted aceto-white and iodine-negative, with different features depending on the lesion grade [24,25].

The study was conducted according to the guidelines of the Declaration of Helsinki and approved by the Ethical Committee of the Emergency University County Hospital of Craiova. This study is a retrospective one, and all patient data has been anonymized. Due to the fact that all patient’s observation papers included written consent for university studies, a separate protocol was not needed.

Based on the cytology and colposcopy tests, the patients accepted in our study were split into those with low-grade lesions and those with high-grade lesions. Depending on the extent of the lesion and the lesion characteristics, several types of interventions were performed: excision of the transformation zone or conization for lesions with moderate or severe dysplasia and targeted cervical biopsies for lesions suggestive of mild dysplasia. The colposcopic guided biopsies have a high accuracy rate regarding the location of the lesions and the cancer-free edges [26].

The colposcopy examination suggested a Low-Grade Dysplasia in 38.63% of the total patients included in this study, and they were re-evaluated after six months. 

The rest of the total patients were suspected at the colposcopy examination of having High-Grade Dysplasia, and they were admitted for an excisional lesion biopsy. In total, 78.60% of these were admitted for an excisional lesion biopsy in NPP and 21.39% in SP. There is a quarterly decrease in the number of cases diagnosed with High-Grade Dysplasia in SP.

Preneoplastic epithelial proliferative lesions were divided according to the affected layers. The patients included in this study were analyzed and centralized by the histopathological diagnosis and the type of treatment applied. The excised fragments were sent to the Pathological Anatomy Laboratory for Histopathological examination. The lesion degrees were established by staining the tissue fragments with hematoxylin-eosin and immunostaining them with the anti-Ki-67 antibody (Table 1) (CIN 1: basal layer was marked, CIN 2: basal layer and intermediate layers were marked, CIN 3: all three epithelial layers were marked: basal, intermediate, and superficial; microinvasive cervical cancer (MCC): the neoplastic cells invade the stroma, in one or more sites, to a depth of ≤3 mm below the base of the epithelium, without lymphatic or blood vessel involvement).

The data used for this study were collected from the Archive of the Obstetrics and Gynecology Clinics of the University Emergency County Hospital of Craiova.

## 3. Results

After analyzing the data, we had 396 patients with pathological results on the PAP smear. All the patients were clinically examined and with colposcopy. In total, 38.63% of them had the aspect of a Low-Grade Dysplasia (Figure 1) on the colposcopy examination and were treated with medication and reevaluated after six months. A total of 61.36% of patients had a High-Grade Dysplasia aspect (Figure 2 and Figure 3) on the colposcopy for which we performed an excisional biopsy of the lesion or Large Loop Excision of the Transformation Zone (LEETZ). Of these patients, 78.60% underwent a biopsy or LEETZ in NPP and 21.39% in SP. There is a decrease of approximately 75% in cases diagnosed with High-Grade Dysplasia in SP. 

In the year before the SARS-CoV-2 pandemic, 78.60% of patients with pathological results of the PAP smear and colposcopy were admitted to the OG Clinics (34.15% patients—OG I Clinic, 44.44% patients—OG II clinic). During the pandemic, only 21.39% of total patients were admitted to the OG clinics with pathological results on the PAP smear and colposcopy (8.64% patients in the OG I clinic and 12.75% patients in the OG II clinic) (Figure 4). There is a significant decrease of 57.21% in the number of cases diagnosed with High-Grade Dysplasia in SP. 

All the patients admitted to the OG I/II Clinics were treated surgically with excisional biopsy or LEETZ. We noticed that the number of surgical procedures had lowered during the first year of the SARS-CoV-2 pandemic. Of the total patients who had been diagnosed with High-Grade Dysplasia before, 78.60% were treated surgically in NPP and 21.39% in SP. In NPP, we realized 33.74% of the total surgical procedures were in OG I, and 44.85% of the total surgical procedures were in OG II. In SP, 21.39% of the total surgical procedures were treated surgically. During the SP, we realize 8.23% of the total surgical procedures were in OG I and 13.16% of the total SP surgical procedures were in OG II (Figure 4). During the NPP in the OG Clinics, excisional biopsies were performed in 31.27% and LEETZ in 47.32% of the total surgical procedures performed. In OG I Clinic, we performed excisional biopsy in 12.75% and LEETZ in 20.98% of the total surgical procedures performed during the NPP. In the OG II Clinic, we performed in NPP excisional biopsy in 18.51% of total surgical procedures performed and LEETZ in 26.33% of the total surgical procedures performed. During the SP in the OG Clinic, excisional biopsy was performed in 6.17% and LEETZ in 15.22% of the total surgical procedures performed. In the OG I Clinic, we performed excisional biopsy in 2.46% and LEETZ in 5.76% of the surgical procedures performed in SP. In the OG II Clinic, we performed excisional biopsy in 3.70% and LEETZ in 9.46% of the total surgical procedures performed in SP (Figure 5). We observed a significantly lower rate of diagnosis and treatment of preneoplastic/neoplastic lesions of the cervix during SP compared with NPP. 

Comparing the type of procedures performed in each month, in both periods, NPP or SP, we observed an important decrease in the number of procedures, such as LEETZ or excisional biopsy (EB): in March of SP, the percentage of LEETZ interventions decreased to 5.75%, and the EB type to 4.52%; in April of SP, the percentage of LEETZ interventions decreased with 3.29%, and the EB type with 2.88%; in May of SP, the percentage of LEETZ interventions decreased to 5.76% and the EB type with 4.52%; in June of SP, the percentage of LEETZ interventions decreased to 5.76%, and that of EB to 2.88%; in July of SP, the percentage of LEETZ interventions decreased to 2.47%, and that of EB to 3.29%; in August of SP, the percentage of LEETZ interventions maintained at 0.82%, and the EB decreased to 0.41%; in September of SP, the percentage of LEETZ interventions decreased to 0.82%, and that of EB to 1.23%; in October of SP, the percentage of LEETZ interventions decreased to 0.83%, and that of EB to 1.64%; in November of SP, the percentage of LEETZ interventions decreased to 2.47%, and the EB to 0.82%; in December of SP, the percentage of LEETZ interventions decreased to 1.64%, and the EB remained at 0.82%; in January of SP, the percentage of LEETZ interventions decreased to 1.64%, and the EB to 0.41%; and in February of SP, the percentage of LEETZ interventions decreased to 2.06%, and that of EB to 2.47% (Figure 6 and Figure 7).

After performing the excisional biopsy or LEETZ for the 243 patients, the fragments of tissue excised were sent to the Pathological Anatomy Service for histopathological examination. The fragments from the excised tissues were examined using classical staining techniques, such as hematoxylin-eosin and immunohistochemical staining with Ki-67 antibodies to identify the lesion degree (CIN1/2/3). In CIN 1, the basal layer was stained; in CIN2, the basal layer and intermediate layers were stained; in CIN3, all layers—basal layer, intermediate layers, and superficial layers—of the epithelium were stained, and in MCC, the neoplastic cells crossed the basement membrane (Figure 8 and Figure 9).

The fragments biopsied were diagnosed with dysplastic lesions. During the NPP in OG Clinics, we identified 29.62% new patients with CIN 1 (OG I—13.58% patients, OG II—16.04% patients), 26.74% patients with CIN 2 (OG I—11.52% patients, OG II—15.22% patients), 15.22% patients with CIN 3 (OG I—5.76% patients, OG II—9.46% patients), and 6.99% patients with MCC (OG I—2.88% patients, OG II—4.11% patients). During the SP, the number of patients diagnosed with cervical intraepithelial neoplasia considerably lowered because of the decreasing number of admissions and surgical procedures during the pandemic. In the OG Clinics, we diagnosed 10.69% of new patients with CIN 1 (OG I—3.7% patients, OG II—6.99% patients), 5.76% of patients with CIN 2 (OG I—2.46% patients, OG II—3.29% patients), 3.29% of patients with CIN 3 (OG I—1.23% patients, OG II—2.05% patients), and 1.64% of patients with MCC (OG I—0.82% patients, OG II—0.82% patients) (Figure 10 and Figure 11). There is a decrease in MS in almost a quarter of the number of cases histopathologically diagnosed compared to NPP.

## 4. Discussion

### 4.1. Context of the Research, Diagnosis Results Following Screening

Romania has a very high incidence of cervical cancer. In Romania, in 2012, the number of newly registered cases was 28.6 per 100,000 inhabitants, and the number of deaths per year was 10.8 per 100,000 inhabitants. However, cervical cancer can be prevented by both primary and secondary prevention. Primary prevention refers to vaccinating girls with the human papillomavirus (HPV) vaccine (preferably before the beginning of their sexual life). Secondary prevention refers to the application of an effective screening program in the detection and treatment of precancerous lesions [27].

According to GLOBOCAN 2020, Romania registered 3380 new cases of cervical cancer this year, representing 7.5% of the total declared new neoplasms [28]

According to other studies, in Romania, the cervical cancer incidence is 34.9/100,000 women, with a total of 4343 new cases per year. The motivation of the research was triggered by the unfortunate situation of Romania, one of the most affected European countries, with an incidence 3.5 times higher than the average in the European Union [27].

The mortality rate of this condition has wide variations in the European Union, with the highest values in Romania—14.2 women annually. In Romania, the mortality rate is approximately 1909 deaths/year. Compared to our country, Iceland has a 20 times lower mortality rate. Romania’s mortality rate is four times higher compared to the average rate in the European Union (3.5/1000) [29].

Although cervical cancer is one of the malignancies easy to prevent, in 2012, GLOBOCAN approximated 58,300 new diagnosed cases in Europe, with an incidence rate of 13.4 per 100,000 women and 24,400 deaths, reaching a 4.9% mortality rate [27].

The Romanian Ministry of Health understood the positive impact that a free national screening program could have on reducing the incidence rate and mortality of cervical cancer. They initiated the first national program for cervical cancer screening in 2012–2017 [30]. Unfortunately, due to the difficulties in the countryside, including a lack of media campaigns, only a small percentage of general physicians involved in this program, i.e., 60%, and only 48% involved in the active promotion, reduced funding, the lack of program control and quantification of the women tested, the initial goal of testing 6 million women aged 25 to 64, who did not have a confirmed diagnosis of cervical cancer, failed. At the end of the program, approximately 260,000 screening tests were performed [31].

### 4.2. Influence of the Pandemic Crisis

The addressability of patients diagnosed with cervical cytological abnormalities was much lower during SP.

Regarding the excisional interventions, we noticed that their number decreased 3.6 times in SP. Excisional biopsy procedures were reduced by approximately 5 times, and LEETZ interventions were reduced by approximately 3.1 times in SP. We observed a significantly lower rate of diagnosis and treatment of preneoplastic/neoplastic lesions of the cervix during SP compared with NPP.

Analyzing the months from NPP and those from SP, we noticed that the biggest decrease for LEETZ in SP was registered in May and June (5.75%) and in those of EB type in the months’ March and May (4.52%). The smallest decreases in the cases of LEETZ in SP were registered in August (0%) and September (0.82%), and the EB type ones in December (0%), August (0.41%), and January (0.41%). These minimal decreases can be caused by the specific period of vacation. Therefore, we can say that both the lack of presentation of patients to the doctor and the lack of medical staff in medical institutions can lead to the decimation of the number of cases treated and diagnosed with preneoplastic lesions of the cervix.

Most patients included in our study had mild dysplastic lesions—CIN 1 (29.62% in NPP and 10.69% during SP). CIN 2 was present in a lower percentage (26.74% in NPP and 5.75% in SP), CIN 3 (15.22% in NPP and 3.29% in SP), and MCC in a minimum percentage (2.88% in NPP and 1.64% in SP). However, the severity of the histopathological lesions was similar, regardless of the period covered by the study. There is a significant decrease in percentage between the two periods. The diagnosis rate of CIN 1 decreased to about one-third in SP. The diagnosis rate of CIN 2 decreased by about 80%, that of CIN 3 also decreased by about 80%, and the diagnosis of MCC was halved in SP.

This may be due to a decrease in cervical screening addressability or due to a tendency to access the medical private system, both explained by the fear of not being infected with the SARS-CoV-2 virus.

The COVID-19 pandemic has strongly influenced the screening tests for dysplastic lesions and cervical cancer worldwide.

The beginning of the pandemic caused a break in cervical screening services. We expect an increased diagnosis rate of advanced lesions of cervical cancer and deaths in the upcoming years [32,33,34,35].

In England, the screening for this pathology was temporarily interrupted from April to June 2020. As a primary care measure, a 6-month re-evaluation was available in case of need [33,34]. Comparable data were recorded in other countries [36,37,38,39,40]. The pandemic waves brought new interruptions of screening for dysplastic and malignant cervical lesions.

Currently, the pandemic is not over. The primary care system and laboratory studies must adapt to restore cervical screening services during the pandemic, providing support even for the waiting patients that were not investigated during this period. However, patients must overcome the fear of seeing a doctor in a pandemic and accept medical evaluation during the SARS-CoV-2 pandemic. The incidence of cervical cancer is expected to increase if compensatory screening programs during the pandemic period do not resume. Assuming that healthcare services will not be able to increase the screening rate for cervical lesions at a level comparable with the previous years or even higher, it will become impossible to reduce the incidence of cervical cancer. Therefore, there will be a delay in the diagnosis of cervical cancer for the women undergoing a screening (25–64 years) according to the standard international protocols [41].

The impact of COVID-19 on cervical cancer monitoring will be difficult to assess if the global impact expands over several years. Furthermore, introducing a primary screening for HPV in 2020 would have led to an initial increase in the diagnoses of premalignant or malignant lesions of the cervix because the primary screening test is more sensitive than exfoliative cytology [42].

Diagnostic delays in malignant or dysplastic cervical lesions will only be visible when we evaluate the screening histories of the patients diagnosed with cervical cancer after resuming the cervical cancer screening programs.

Currently, there are some attempts to resume screening for cervical cancer. There are still issues that disrupt primary, secondary, or tertiary care. To solve these problems, we try to approach the strategies of recovery of screening programs despite the persistence of COVID-19 by finding recommendations (target age, periodic screenings, clinical follow-up after finding positive results for intraepithelial lesions) and minimizing the risk, with an impact on resources, costs, and quality of life. There are many obstacles in maximizing the results, including the medical staff, the diagnostic equipment (used also for detecting the SARS-CoV), and the budget (used mostly for the diagnose of the SARS-CoV virus during the pandemic) [43].

Following this study, we highlighted that the SARS-CoV-2 pandemic has left marks on the number of patients hospitalized and diagnosed with CIN, so in the next period, there may be a greater number of patients with high-grade dysplastic and invasive lesions due to the lack of medical care by decreasing the number of hospitalizations of patients in hospitals and implicitly by decreasing curative surgical maneuvers.

Similar decreases in the number of cervical screening tests during the COVID-19 pandemic have been observed internationally [44,45].

Furthermore, it would certainly be important to know the impact of reducing the diagnosis of CIN lesions on the incidence and diagnosis of cervical cancer after the COVID pandemic, and this is one of the long-term goals set by our center.

## 5. Conclusions

The premalignant and dysplastic lesions are easy to diagnose and treat if investigated in time. During the SARS-CoV-2 pandemic, we found almost a four-time lower rate of patient admissions, diagnosis, and treatment. Considering that, the number of patients left undiagnosed and untreated can be much higher due to restrictions in the healthcare system generated by this pandemic or the fear of being infected with the SARS-CoV-2 virus. The lower rate seems to be a direct result of undiagnosed and untreated cases due to the restrictions in Romania’s healthcare systems as a result of the SARS-CoV-2 pandemic.

## Figures and Tables

**Figure 1 medicina-57-01101-f001:**
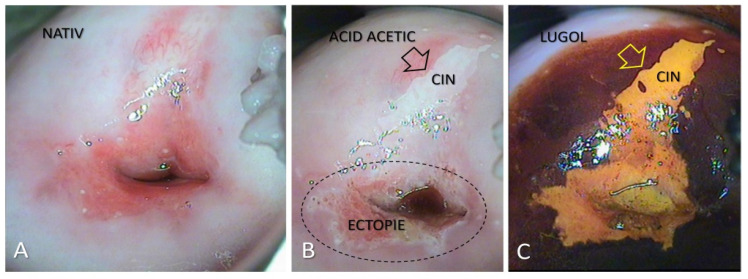
Colposcopic examination showing CIN 1 cervical lesion. (**A**): After the application of saline, native appearance; (**B**): After the application of acetic acid: Fine, bright-white acetone-white epithelium (arrow), visible; (**C**): After applying the Lugol solution: iodine-negative lesion, with well-defined edges.

**Figure 2 medicina-57-01101-f002:**
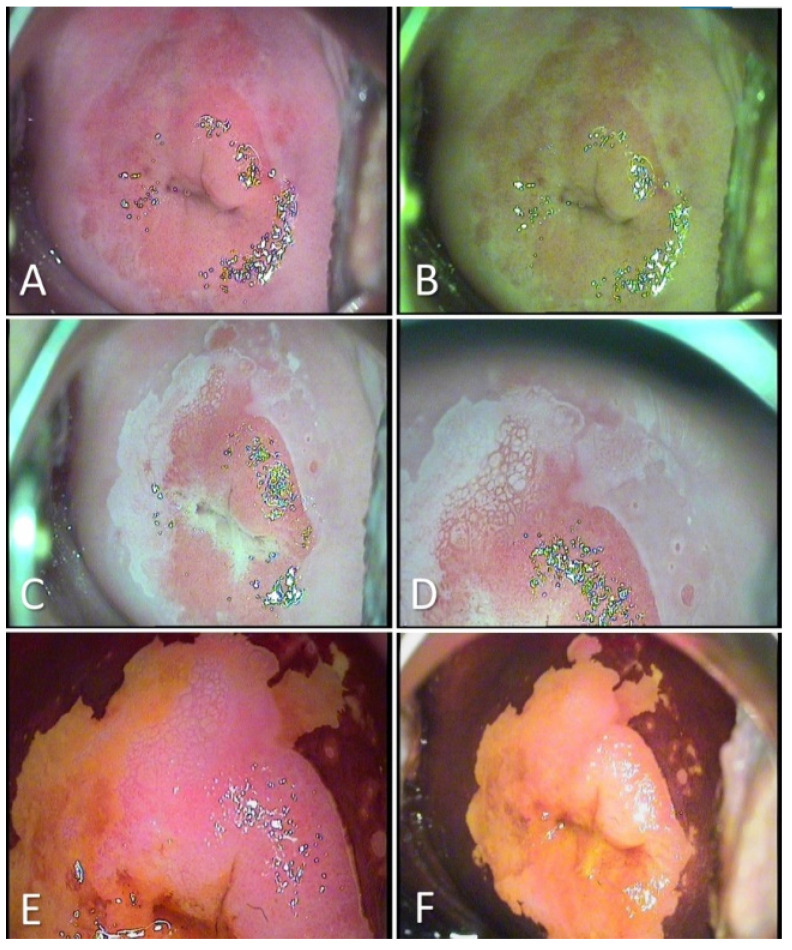
Colposcopic examination showing HSIL/CIN 2/3 cervical lesions. (**A**): After the application of saline: Native appearance, no visible lesion; (**B**): Evaluation in green light filter: does not highlight abnormal vessels; (**C**,**D**): After applying acetic acid: Aceto-white epithelium elevated, bright-white, and mosaic visible in three quadrants of the cervix; (**E**,**F**): After applying the Lugol solution: iodine-negative lesion, with well-contoured edges and circumferential extension.

**Figure 3 medicina-57-01101-f003:**
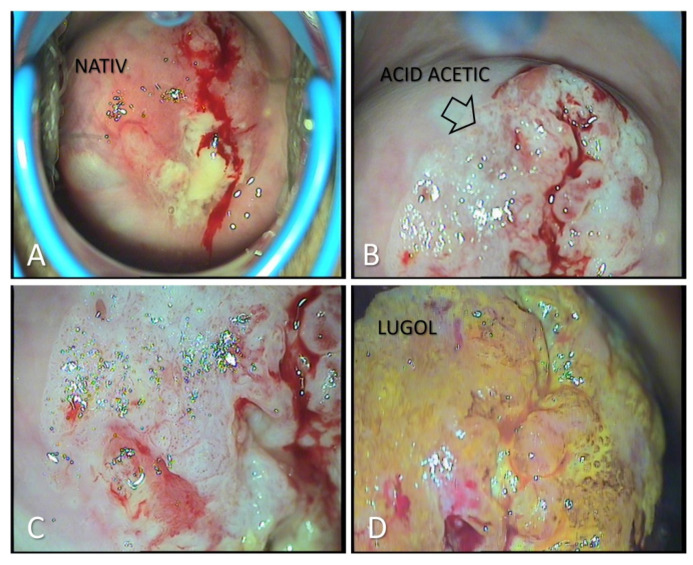
Colposcopic examination pleading for HSIL cervical lesion, with potential invasive carcinoma. (**A**): Native appearance: disorganized structure of the cervix, friable, easy bleeding to the touch; (**B**): After the application of acetic acid: extensive areas of aceto-white epithelium elevated and vessels with an increased caliber and active bleeding; (**C**): After applying acetic acid: association of multiple lesions, coarse lesions, associated with erosions and ulcerations; (**D**): Negative reaction when applying the Lugol solution, with areas of intense yellow coloration, such as “saffron” or “canary”.

**Figure 4 medicina-57-01101-f004:**
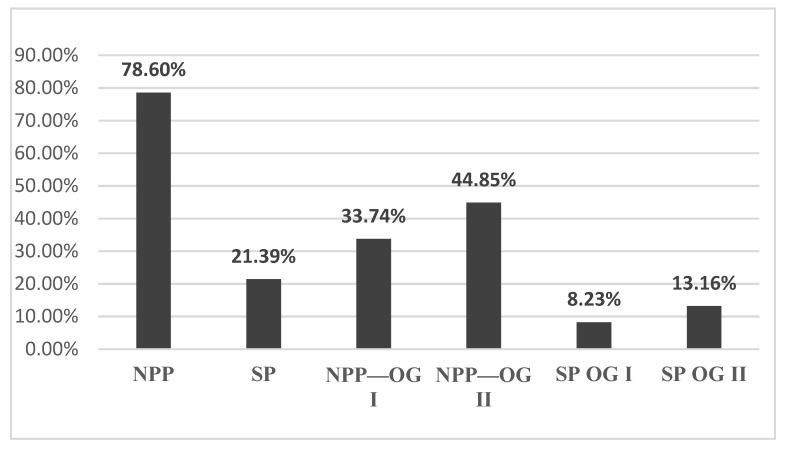
The total number of hospitalized patients diagnosed with cervical lesions by PAP test and colposcopy in the year before the pandemic and the year of the SARS-CoV-2 pandemic. OG: Obstetrics and Gynecology Clinic. NPP: non-pandemic period; SP: SARS-CoV-2 period.

**Figure 5 medicina-57-01101-f005:**
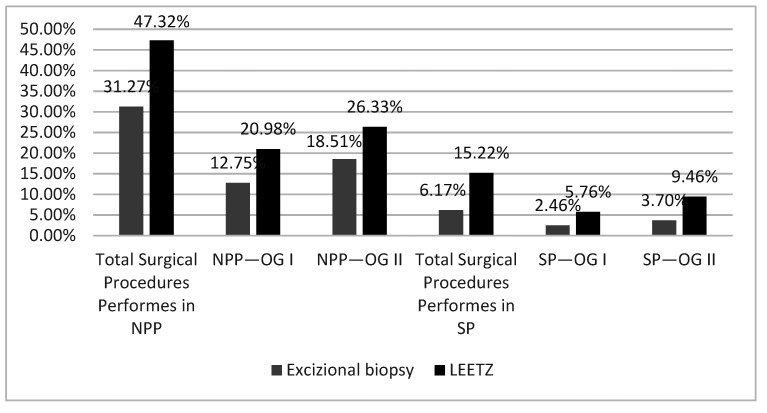
Surgical procedures performed in both clinics of the Emergency County Hospital in the year before the pandemic and the first year of the SARS-CoV-2 pandemic. OG: Obstetrics and Gynecology Clinic. NPP: non-pandemic period; SP: first year of SARS-CoV-2 pandemic; LEETZ: Large loop excision of the transformation zone.

**Figure 6 medicina-57-01101-f006:**
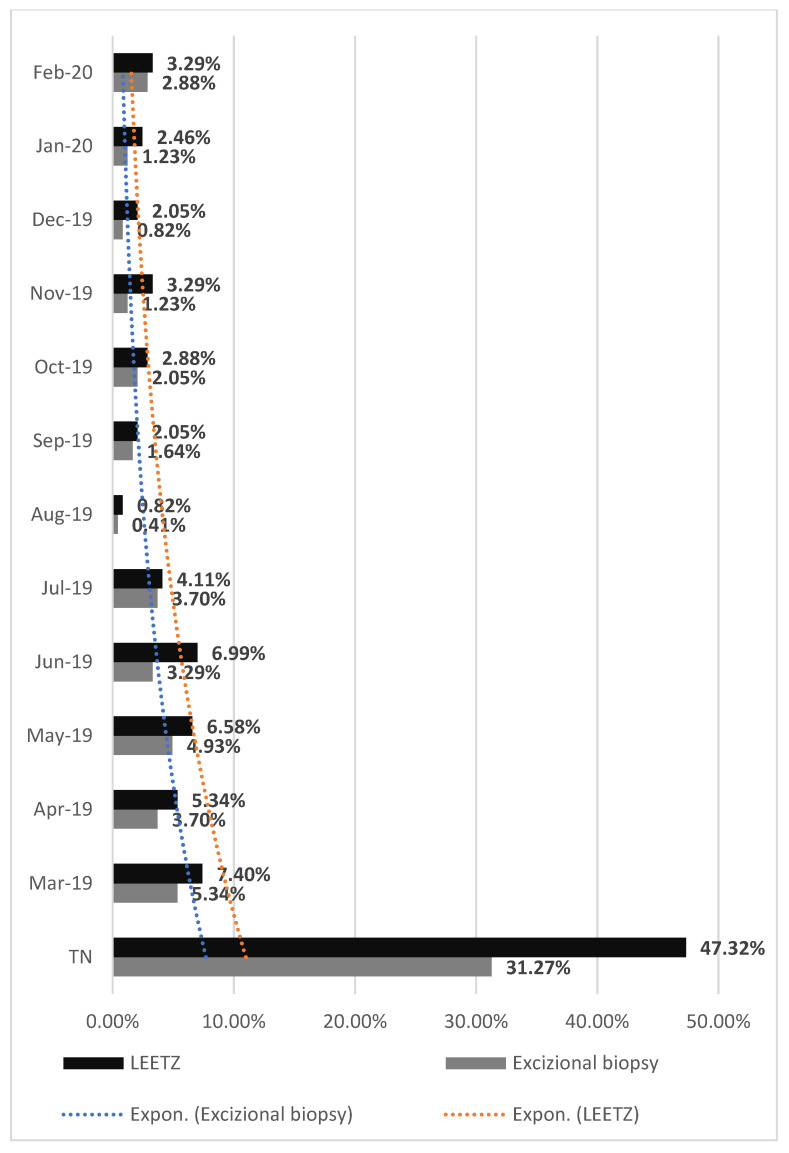
Surgical procedures performed in both clinics of the Emergency County Hospital in the year before the SARS-CoV-2 pandemic. NPP: non-pandemic period; LEETZ: Large loop excision of the transformation zone; TN: Total number of surgical procedures performed.

**Figure 7 medicina-57-01101-f007:**
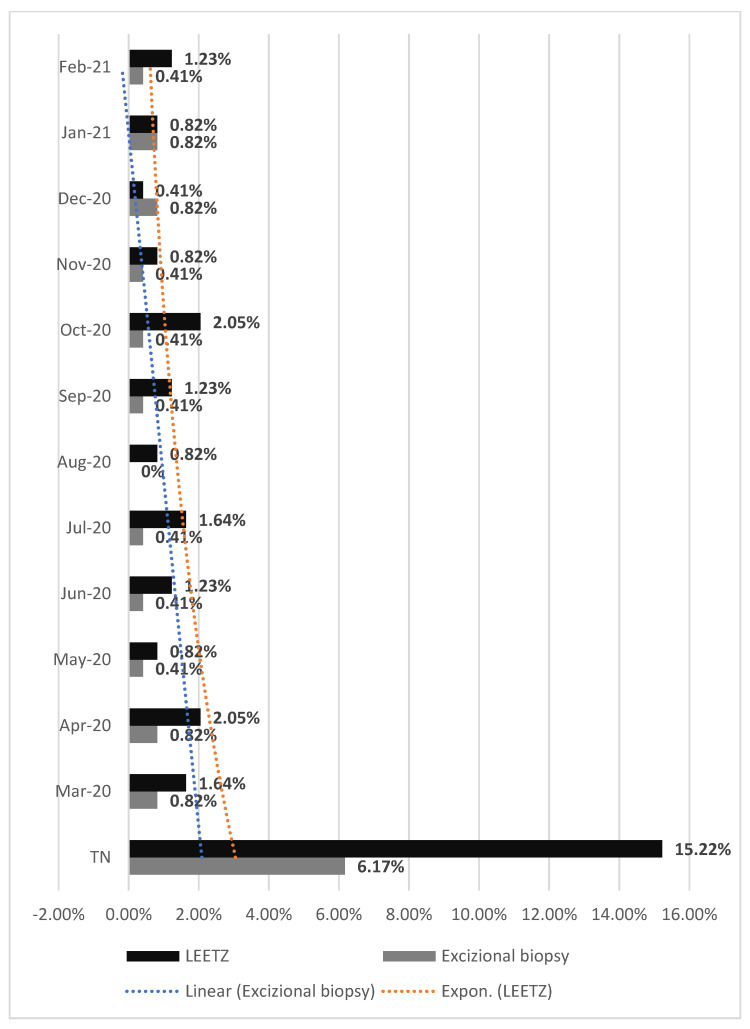
Surgical procedures were performed in both clinics of the Emergency County Hospital in the year of the SARS-CoV-2 pandemic. SP: first year of SARS-CoV-2 pandemic period; LEETZ: Large loop excision of the transformation zone; TN: Total number of surgical procedures performed.

**Figure 8 medicina-57-01101-f008:**
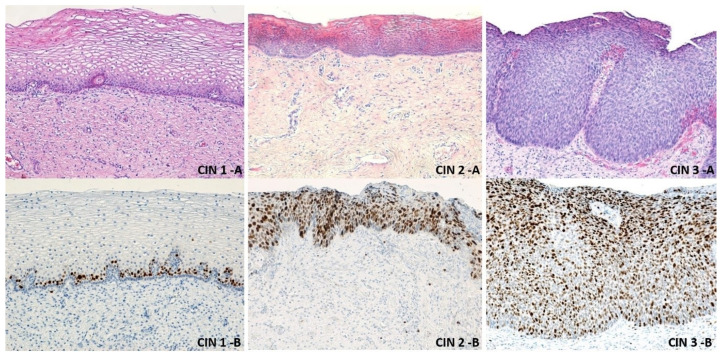
Microscopic aspects of cervical intraepithelial neoplasia. CIN 1-A: non-keratinized stratified squamous epithelium with low-grade dysplastic lesions in the basal layer. Classical stained hematoxylin-eosin × 100; CIN 1-B: non-keratinized stratified squamous epithelium with low-grade dysplastic lesions in the basal layer, with the nucleus of the cells immunostained with the Ki-67 antibody × 100; CIN 2-A: non-keratinized stratified squamous epithelium with middle-grade dysplastic lesions in the basal and intermediate layers. Classical stained hematoxylin-eosin × 100; CIN 2-B: non-keratinized stratified squamous epithelium with middle-grade dysplastic lesions in the basal and intermediate layers, with the nucleus of the cells immunostained with the Ki-67 antibody × 100; CIN 3-A: non-keratinized stratified squamous epithelium with severe dysplastic lesions in all the tissue layers. Classical stained hematoxylin-eosin × 100; CIN 3-B: non-keratinized stratified squamous epithelium with severe dysplastic lesions in all the tissue layers, with the nucleus of the cells immunostained with the Ki-67 antibody × 100; CIN: cervical intraepithelial neoplasia.

**Figure 9 medicina-57-01101-f009:**
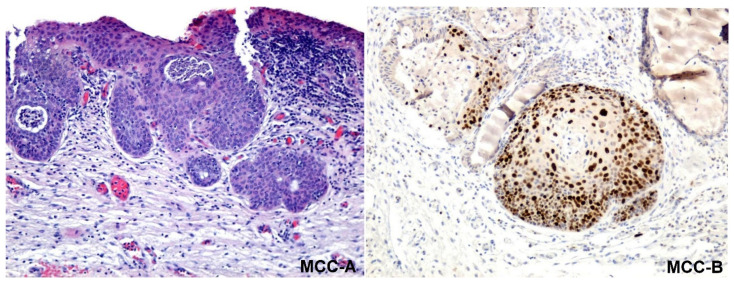
Microscopic aspects of microinvasive cervical cancer. MCC-A: non-keratinized stratified squamous epithelium with severe dysplastic lesions in all the tissue layers and the existence of a microinvasion island beyond the epithelial basement membrane is observed. The neoplastic cells invade the stroma, in one or more sites, to a depth of ≤3 mm below the base of the epithelium, without lymphatic or blood vessel involvement. Classical stained hematoxylin-eosin × 100; MCC-B: the existence of an island of microinvasion immunolabeled with the anti-Ki67 antibody can be observed, which has crossed the epithelial basement membrane. MCC: microinvasive cervical cancer.

**Figure 10 medicina-57-01101-f010:**
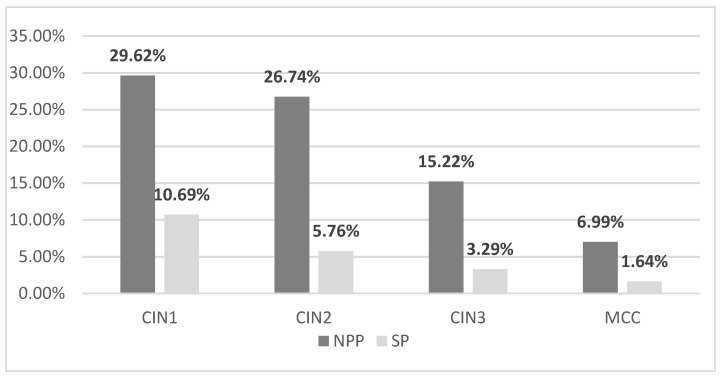
The rate of dysplastic lesions diagnosed using a histopathological examination: the year before the pandemic and the year of the SARS-CoV-2 pandemic in both clinics of the Emergency County Hospital. CIN: cervical intraepithelial neoplasia; MCC: microinvasive cervical cancer; NPP: non-pandemic period; SP: SARS-CoV-2 period.

**Figure 11 medicina-57-01101-f011:**
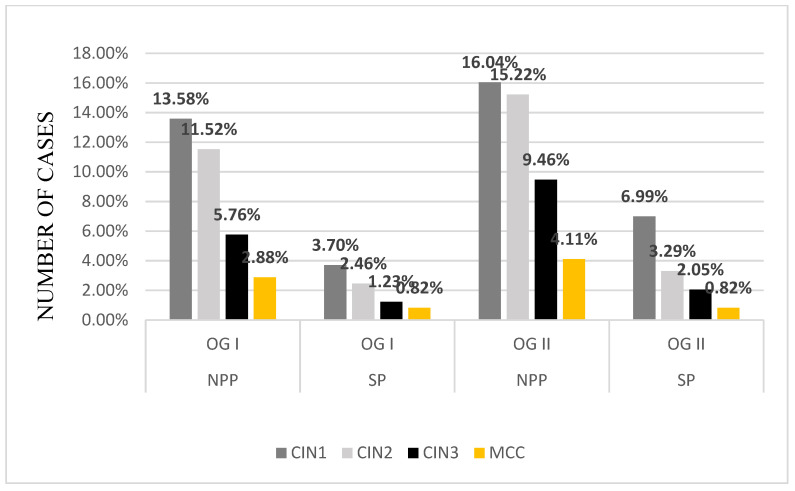
The rate of dysplastic lesions diagnosed using histopathological examination: the year before the pandemic and the first year of the SARS-CoV-2 pandemic. OG: Obstetrics and Gynecology Clinic; CIN: cervical intraepithelial neoplasia; MCC: microinvasive cervical cancer; NPP: non-pandemic period; SP: first year of SARS-CoV-2 pandemic.

**Table 1 medicina-57-01101-t001:** The immunohistochemical panel of antibodies used for immunostaining.

Anti-Ki67 ^1^
Dako
MIB-1
Ethylenediaminetetraacetic acid (EDTA), pH9
Monoclonal mouse anti-human Ki67
1:50
Cells in the division in G1, S, G2 and M phase

^1^ Ki67: Marker of Proliferation Ki-67; G_1:_ Growth 1 phase; S: Synthesis phase; G_2_: Growth 2 phase; M: Mitosis phase.

## Data Availability

Data was collected from the Emergency University County Hospital of Craiova from the Obstetrics-Gynecology Clinics. This data is not available online but can be provided upon formal request.

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
