# Peer review of "The Influence of SARS-CoV-2 Pandemic in the Diagnosis and Treatment of Cervical Dysplasia"

_medicina, 2021, doi:10.3390/medicina57101101_

Round 1
Reviewer 1 Report
Well written article in a very interesting topic. Authors present in details, how COVID pandemia affected patients admission at with cervical pathology. Their result are of great importance, as cervical cancer is very common in Romania.
Author Response
Author's Reply to the Review Report (Reviewer 1)
Author response in red.
Dear Reviewer,
Thank you very much for your comments. They helped us to improve the quality of the manuscript. We made revisions according to your suggestions. Please, find below detailed point-by-point answers to your comments.
- English language and style are fine/minor spell check required
Reply #1: A revision of the English language was performed. All changes are highlighted by using track changes.
Reviewer 2 Report
Thanks for the hard work,
The low presentation and admission is a global phenomenon
This is not a new information
Suggestion:
The better study for major international readership will be '' Incidence and severity of cancer diagnosed of post COVID period and overall long term impact
Reviewer 3 Report
This is a very important topic. The effect on the health of women during and now post-pandemic is of paramount importance. And especially when it comes to a preventable illness. The study is fine, the numbers are fine, conclusions are fine.
The problems I have are with the writing, the references, the statements that don't correlate to references nor are they appropriate. Also use of references from 10 and even 30 years ago to describe existing states of screening/disease numbers makes no sense (see the use of references number 7 and number 8 as examples). Also, reference 9 is it really a study by the WHO? Covid numbers...way off, much more than 2.4 million and 166,000 deaths. Or are you referring to a region? I apologize if the manuscript was written when these were the accurate number of cases, but they should be updated.
The first sentence of the discussion is completely wrong: "Romania has the highest incidence of cervical cancer". I would love to re-read this article after these inconsistencies are fixed. I really think it is an important study, and it is well done, but can't be presented with these errors. There may be other errors, but I skimmed it....just the few I pointed out made me lose my enthusiasm for reading it or trusting the findings.
Round 2
Reviewer 2 Report
Uggestion:
It is good that the Authors have added those texts in the introduction and Discussion paragraphs.
However, I would suggest
DELETE lines 51-66 and 100-110 in the introduction. Readers know about this. It is too lengthy. People start losing interest as it appears like a BOOK CHAPTER or AN ESSAY.
DELETE lines 366 to 398. No need to tell how you diagnose or confirm the diagnosis. This is not the aim of this article and makes this paper too lengthy.
DELETE lines 454-458
The scientific publication should be specific to the study aim and findings. But too much information and discussion make it look like a book chapter ( Clinicians attract to focused and crisp discussion )
Readers lose interest in lengthy discussions and page-long introductions.
Similarly, the Conclusion is a very general comment, try to relate it with your study finding like as below (example)
'' in this study, we found that due to COVID there was generally less access to screening programme which resulted in .......... and future impact can be ....etc etc etc'''
It is a common issue that Authors add too much information as it seems very important and nobody contradicts but the main question is that providing that info is may not add any value to your study and also not giving any new information which wider readership already don't know.
Method and Material is also very very lengthy
Anyhow, I cant see that study like this should be more than 1500-2000 words maximum.
Reviewer 3 Report
Thanks for the thorough review of the comments. There are still additional commas here and there, but the main points come across.
minor suggestions/etc.
line 103: should there be an age? i.e., "...every 3-5 years...(just suggestions) after sexual activity begins/OR lets's say after age 26...or something?
Sentence line 117/118 missing or should start with Protocols have been adopted...
Line 355 "100000" is "1ooooo"
Line 484 (suggestion)Considering that, the number of patients 484 left undiagnosed and untreated can be much high, due to restrictions in the healthcare 485 system generated by this pandemic, or the fear of being infected with the SARS-CoV-2 486 virus. This lower rate is (or "seems to be") a direct result of undiagnosed and untreated cases due to the restriction in Romania's healthcare systems as a result of the SARS-CoV-2 pandemic.
